

# A data-driven artificial neural network approach to predict make-up exam participation in higher education

Ozcan Cataltas

Faculty of Technology, Electrical and Electronic Engineering Department, Selcuk University, Konya, Turkey

## ABSTRACT

In the Turkish higher education system, make-up exams are a key mechanism enabling students to recover failed courses, yet the decision to take the make-up exam depends on multiple student- and course-level factors. This study develops an artificial neural network (ANN) model to predict whether students will take the make-up exam, using routinely available academic and course-related features. To obtain a comprehensive sample reflecting both student achievement and course characteristics, the dataset was drawn from courses spanning different academic levels and difficulty tiers. During data preparation, normalization and one-hot encoding were applied to facilitate model learning. Training, validation, and test subsets were formed using a Stratified Sample Projection in X and y (Stratified SPxy) sampling strategy to maintain representativeness across splits, and hyperparameters were optimized *via* random search to identify the best-performing configuration. The final model achieved 87.68% test accuracy and an 87.62% test F1-score, indicating good generalization. Notably, to the best of our knowledge, this is the first study to address make-up exam participation using only easily accessible institutional data, which makes the proposed approach practical and adaptable for real-world university settings. Despite relying on simple and readily available parameters, the ANN delivers strong predictive performance, suggesting that efficient models can be developed for non-complex datasets. Such predictions can enable early identification of at-risk students and trigger targeted academic support, inform course- and program-level adjustments, and improve operational planning, thereby contributing to more effective educational management and improved student progression and completion rates.

## INTRODUCTION

In recent years, artificial intelligence (AI) and machine learning have demonstrated substantial potential to extract meaningful information from large datasets, make forward-looking predictions, and optimize complex processes across domains such as health (*Topol, 2019*), finance (*Heaton, Polson & Witte, 2018*; *Weinan, Hu & Peng, 2023*), transportation (*Ma et al., 2020*), and education (*Zawacki-Richter et al., 2019*). In health, for

Corresponding author
Ozcan Cataltas,
ozcancataltas@selcuk.edu.tr

example, AI supports disease diagnosis (*Esteva et al., 2017*) and treatment planning (*Jiang et al., 2017*), while in finance it aids fraud detection and risk assessment (*West & Bhattacharya, 2016*). The recent emergence of generative AI further broadens educational applications; however, the present study focuses on supervised prediction for decision support rather than on generative models, artificial general intelligence, or superintelligence (*Monib et al., 2024*).

Education is a prominent application area where AI and machine learning enable data-informed decisions and operational efficiency (*Holmes, Bialik & Fadel, 2019*). Machine learning methods are widely used for predicting student achievement (*Kotsiantis, Pierrakeas & Pintelas, 2004*), enhancing e-learning technologies (*Mahafdah, Bouallegue & Bouallegue, 2024*), personalizing learning processes (*Baker, Martin & Rossi, 2016*), and optimizing educational resources (*Lin & Albahli, 2025*). By enabling accurate predictions and advanced analytics, these methods help educators better understand students' needs, tailor support, and improve academic outcomes (*Ma et al., 2014*; *Romero & Ventura, 2024*).

The Turkish university system evaluates students primarily through midterm and final exams, typically combined with a 40–60% weighting to determine end-of-semester achievement. Some universities adopt relative grading based on cohort performance, while others apply absolute grading based on individual performance. For students who do not pass, make-up exams replace the final and offer an additional opportunity to pass the course. Make-up exams, therefore, play a critical role for both institutions and students, especially those nearing graduation. Students' decisions to take a make-up exam, however, depend on multiple factors, including individual motivation, perceived exam difficulty, midterm performance, number of failed courses, distance from campus or hometown, academic status, and course difficulty, which introduces uncertainty for planning (*Soghier & Qu, 2013*; *Gür & Köroğlu, 2023*).

Accurately anticipating participation in make-up exams matters for more than logistics. Predictions can support timely, targeted interventions for students likely to need an additional assessment opportunity, which has been associated with improved learning outcomes and course completion in data-driven educational settings (*Romero & Ventura, 2024*). At the program level, cohort-level risk signals can guide assessment design, resource allocation, and the provision of supplemental sessions in courses with higher make-up demand. Operationally, better forecasts can also improve scheduling of rooms and invigilators and avoid unnecessary exam printing, thereby supporting efficient use of institutional resources (*Kaddoura, Popescu & Hemanth, 2022*; *Elbourhamy, 2025*). These operational and environmental benefits are treated as secondary advantages that complement the primary educational value of early support.

Machine learning methods—and artificial neural networks (ANNs) in particular—have been effective in educational prediction tasks. Prior work has used ANN and other techniques to predict academic achievement and analyze exam outcomes (*Basheer & Hajmeer, 2000*; *Kotsiantis, Pierrakeas & Pintelas, 2004*; *Yadav & Pal, 2012*; *Al-Barrak & Al-Razgan, 2016*). More recent studies illustrate both the methodological breadth of contemporary machine learning research and its concrete value for educational decision

making. For example, *Yousafzai et al. (2021)* proposed "Student-Performulator", an attention-based Bidirectional Long Short-Term Memory (BiLSTM) augmented with a Chi-square feature-selection stage; trained on Pakistani higher-education records, the hybrid network lifted grade-prediction accuracy to 90.2%, underscoring the benefit of combining shallow statistical filtering with deep sequence modeling for noisy, categorical student data. Moving one step earlier in the learner lifecycle, *Abideen et al. (2023)* analyzed five years of secondary-school enrolment logs from 100 public schools in Punjab. Using random-forest and decision-tree ensembles, they achieved $R^2 \approx 0.97$ for next-year enrolment counts and—critically—translated the predictions into actionable "below/far-from/on-target" categories for school administrators, illustrating how interpretable tree-based models can support province-level resource allocation. Although situated outside education, *Ahmad et al. (2025)* introduced a Transformer-based multimodal encoder that fuses large-language-model representations with domain-specific clinical signals to improve automated health-monitoring F1-scores from 0.80 to 0.83. However, studies directly predicting participation in make-up exams are limited. To the best of our knowledge, only two studies have addressed this problem: one used k-Extreme Learning Machine optimized with an Artificial Bee Colony algorithm on a dataset from Konya Technical University comprising features such as semester average, grade point average (GPA), number of courses taken, midterm and final scores, number of retaken courses, semester, distance to Konya, gender, and count of failed make-ups (*Kiran, Siramkaya & Eşme, 2021*). The other trained ANNs with different optimization strategies (backpropagation, gray wolf optimization, and random weight initialization) on the same dataset, reporting differing advantages in training accuracy, generalization, and computational efficiency (*Kiran et al., 2022*).

This study contributes to the literature by developing an ANN-based predictor of make-up exam participation using a new dataset built from routinely accessible institutional records. The design emphasizes parsimony and portability: the model relies on fewer, easily obtainable academic and course-level features that universities can extract directly from student information systems, without requiring sensitive or hard-to-collect data. In assembling the dataset, priority was given to variables available at scale in operational settings; nevertheless, this choice may limit breadth and introduce constraints on generalizability beyond the source institution. Moreover, because the focus is on at-risk students, already passing students were excluded when modeling participation, which is pragmatic for the intended use case but may induce selection bias; these limitations are acknowledged and discussed in the Discussion and Limitations section.

Methodologically, data are preprocessed using normalization and one-hot encoding to facilitate model learning and ensure compatibility across heterogeneous feature types. Stratified train/validation/test partitions are formed using the SPxy sampling strategy to maintain representativeness, and hyperparameters are optimized *via* random search over a broad configuration space. To contextualize performance, standard metrics (accuracy, precision, recall, F1, specificity) are reported, and their practical meaning for

educational decision making is discussed. The results are also positioned relative to prior studies.

The remainder of the article is organized as follows. The dataset and preprocessing steps are first described. The modeling approach and training procedure are then presented. Next, results and comparative analyses are reported, followed by a discussion of implications, limitations, and ethical considerations, including generalizability beyond the source university and potential integration into institutional workflows. Finally, contributions are summarized and directions for future research are outlined.

## MATERIALS AND METHODS

### Dataset description

The dataset used in this study comprises records from courses offered by the Department of Electrical and Electronic Engineering at Selcuk University during the Fall semester of the 2022–2023 academic year. Only course and assessment level fields available to administrators were used; no personally identifiable information was accessed. All records were anonymized prior to analysis, and course names were label-encoded (Course1, Course2, …). The study relies on routinely collected administrative data and involved no intervention.

To ensure a representative and comprehensive sample, courses spanning different academic levels and difficulty tiers were selected, and all students enrolled in those courses were included. In line with university regulations, students with passing letter grades (AA–CC)—who are ineligible for the make-up exam—were excluded so that the model focuses on the at-risk population. This pragmatic restriction may introduce selection bias; implications for generalizability are discussed later.

The final dataset contains 1,194 samples, each representing a student–course observation with associated performance indicators and exam status. It includes numerical and categorical features that capture both individual academic achievement and basic course characteristics. No missing data were present, so neither imputation nor case-wise deletion was required.

*Inputs:*

(1) Midterm Score (MS): Student's score in the midterm exam (numerical).

(2) Final Score (FS): Student's score in the final exam (numerical). In the studied setting, students decide on make-up participation after final grades are released; FS is therefore available at decision time.

(3) Weighted Average (WA): Weighted average of midterm and final scores, WA = 0.4 × MS + 0.6 × FS (numerical). Although WA is a linear function of MS and FS, it was retained because it is the official course performance indicator used for letter grading at decision time and can aid institutional interpretability.

(4) Letter Grade (LG): Semester letter grade (*e.g.*, F, FF, DD, DC) before any make-up exam (categorical).

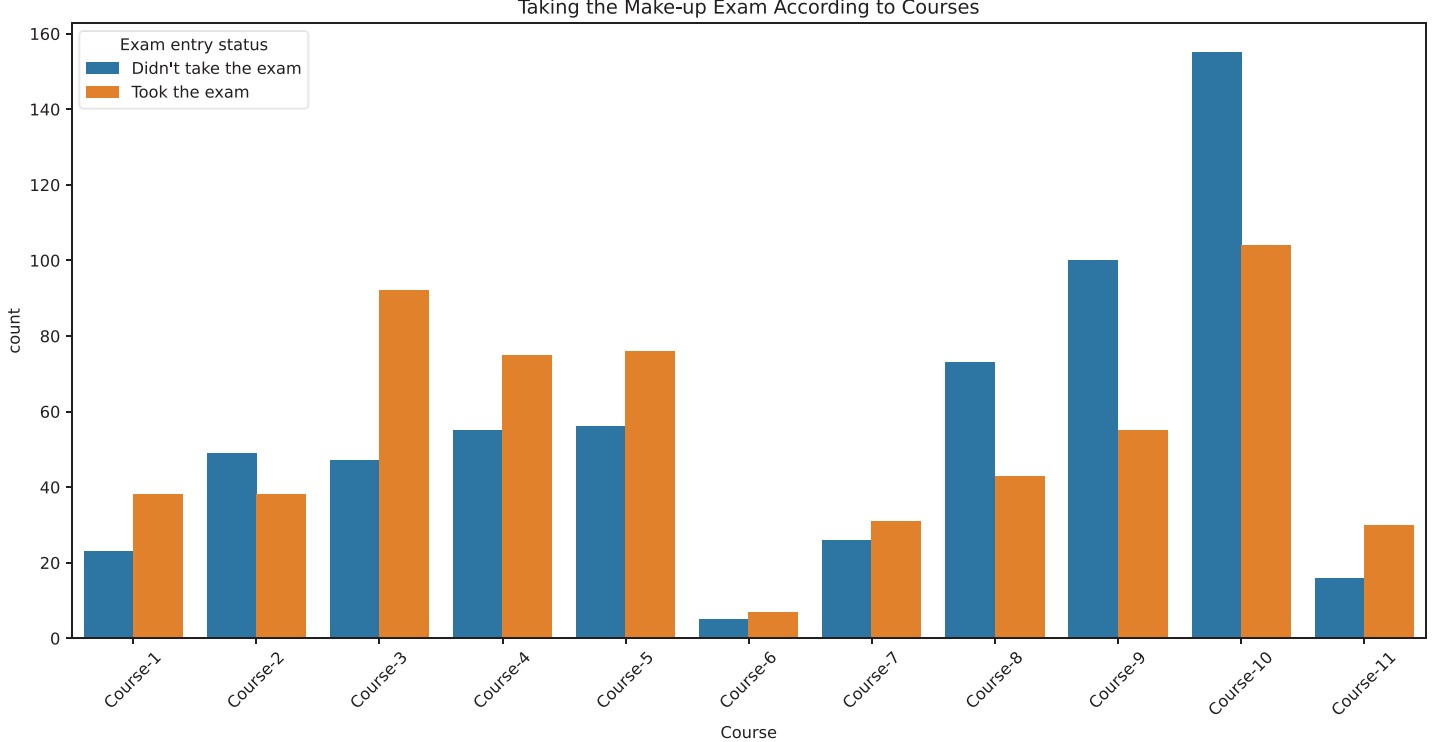

**Figure 1** **Make-up exam participation by course.** Bar chart showing the number of students who took or did not take the make-up exam across different courses. The x-axis represents the courses, while the y-axis indicates the number of students.

(5) Type of Program (ToP): Education program (categorical: first education or second education).

(6) Course Grade Level (CGL): Course level in the curriculum (*e.g.*, 1st, 2nd, 3rd, 4th year) (categorical).

(7) Course Name (CN): Course identifier (labeled as Course1, Course2, ...) (categorical).

(8) Fail Rate (FR): Course level proportion of students eligible for the make-up exam in the given term, computed as (number eligible for make-up)/(total students enrolled in the course) (numerical).

*Output:*

Make-up exam participation: Binary indicator of whether the student took the make-up exam in that course (1 = yes, 0 = no).

These characteristics provide information at both the individual and course levels so that the model can learn from within-course variation as well as systematic differences across courses.

Figure 1 displays, for each course, the counts of students who did and did not take the make-up exam to visualize class balance across courses. Table 1 summarizes dataset characteristics. The mean midterm and final scores were 23.2 and 22.1, respectively, with standard deviations indicating substantial variability. Most students received FF ($n = 948$), while other failing grades were less frequent. The dataset includes both program types in

**Table 1 Descriptive statistics and feature overview of the dataset.**

| No | Feature | Type | Min | Max | Avg ± std | Count |
|---|---|---|---|---|---|---|
| 1 | Midterm score | Numeric | 0 | 100 | 23.2 ± 17.5 | |
| 2 | Final score | Numeric | 0 | 80 | 22.1 ± 18.7 | |
| 3 | Weighted average score | Numeric | 0 | 58 | 22.2 ± 15.2 | |
| 4 | Letter grade | Categorical | | | | F: 85<br>FF: 948<br>DD: 41<br>DC: 120 |
| 5 | Type of program | Categorical | | | | First: 527<br>Second: 667 |
| 6 | Course grade level | Categorical | | | | 1st grade: 358<br>2nd grade: 521<br>3rd grade: 200<br>4th grade: 115 |
| 7 | Course name | Categorical | | | | Course-1: 61<br>Course-2: 87<br>Course-3: 139<br>Course-4: 130<br>Course-5: 132<br>Course-6: 12<br>Course-7: 57<br>Course-8: 116<br>Course-9: 155<br>Course-10: 259<br>Course-11: 46 |
| 8 | Fail rate | Numeric | 0.08 | 0.97 | 0.65 ± 0.2 | |
| 9 | Exam entry status | Categorical | | | | Took the exam: 605<br>Didn't take the exam: 589 |

**Note:**
The features used in the analysis, including their types (numeric or categorical), minimum and maximum values, averages with standard deviations (for numeric features), and counts or distributions (for categorical features).

substantial numbers (first education: 527; second education: 667). The mean course-level fail rate is high (mean FR ≈ 0.70). Overall, the target variable is approximately balanced (took: 605; did not take: 589), which supports reliable estimation without additional rebalancing.

## Data preprocessing

Data preprocessing is a critical step for accurate model training and evaluation. In this study, a small set of deterministic transformations was applied to represent the dataset's characteristics more effectively and to improve learning stability.

First, numeric score variables (Midterm, Final, Weighted Average) were normalized *via* linear rescaling from the native 0–100 range to [0, 1]. Because this mapping is fixed and does not estimate parameters from the data, it cannot introduce train–test leakage. Bringing features to a comparable scale stabilizes gradient-based optimization, reduces the dominance of large range variables on the loss surface, and typically yields faster, more reliable convergence for Rectified Linear Unit (ReLU)-activated networks. Normalized

inputs also help dropout regularization operate more consistently by keeping activations on comparable scales across layers.

Second, in accordance with university regulations, records with successful letter grades (AA, BA, BB, CB, CC)—who are ineligible to take the make-up exam—were excluded so that the modeling focuses on the at-risk population. This exclusion criterion aligns the prediction task with the real decision context. The potential for selection bias due to this restriction is acknowledged, and implications are discussed in the Limitations section.

Third, categorical variables were encoded into a numerical format suitable for machine learning. One-hot encoding was applied to Letter Grade, Type of Program, Course Name, and Course Grade Level to avoid imposing spurious ordinality among categories (*e.g.*, DC > DD > FF), which could distort distance-based representations and the learned decision boundary. One-hot encoding expands each categorical feature into as many binary columns as there are categories, preserving categorical semantics (*Bramwell, 2001*). Although this increases dimensionality and sparsity, the network's layer widths and dropout were selected to accommodate the expanded input space and mitigate overfitting risk.

No missing data were present in the dataset; therefore, no imputation or case-wise deletion was required. All preprocessing steps were applied uniformly to the training, validation, and test splits using the same deterministic procedures.

## Splitting the dataset

To accurately evaluate model performance, the dataset was divided into training, validation, and test subsets using a strictly held-out test set. Specifically, 70% of the data was allocated to training, 15% to validation, and 15% to testing. To preserve both the diversity of input features and the class distribution of the target, a stratified SPxy strategy was employed, which maintains class proportions while encouraging representativeness in feature space (*He & Garcia, 2009*; *Li et al., 2021*).

The SPxy procedure considers relationships in both the input feature space $X$ and the target $Y$. Pairwise distances among samples in $X$ were computed using the Euclidean metric on the normalized inputs:

$$d\left(x_i, x_j\right) = \sqrt{\sum_{k=1}^{n} \left(x_{i,k} - x_{j,k}\right)^2},$$

where $d\left(x_i, x_j\right)$ is the distance between samples $x_i$ and $x_j$. To retain the marginal distribution of the binary target in each subset, the class proportions were preserved based on

$$p_c = \frac{n_c}{N},$$

where $p_c$ is the proportion of class $c$, $n_c$ is the number of samples in class $c$, and $N$ is the total number of samples. Stratification ensures these proportions are approximately maintained in the training, validation, and test splits.

To further promote representativeness with respect to X–y relationships, a projection-based sorting akin to SPxy was used. Let $X$ denote the matrix of input features and $Y$ the target vector; a projection is then formed:

$$P = X^T Y,$$

which provides an ordering that reflects joint structure in $X$ and $Y$. The ordered samples are then assigned to train/validation/test in a stratified manner so that both the class balance and diversity in feature space are preserved across subsets.

We did not employ k-fold or nested cross-validation in order to (i) maintain a single, clean test benchmark free from potential hyperparameter leakage, and (ii) keep computation tractable given the random-search optimization over a broad configuration space. It is acknowledged that cross-validation can further stabilize performance estimates; this trade-off is discussed in the Discussion and Limitations section. All splitting was performed before any model training, with a fixed random seed to ensure reproducibility. Because normalization was a fixed 0–1 rescaling and categorical encodings were deterministic, preprocessing did not introduce train–test leakage.

Table 2 reports the distribution of the dataset across the training, validation, and test subsets. Feature distributions and target classes are preserved across splits. The number of students who took and did not take the make-up exam is nearly equal in each subset. Letter grades, program types, grade levels, and course representation are also consistent across all splits, indicating that the model was trained and evaluated on representative samples.

## Performance metrics

In this study, the performance of the ANN model was evaluated using usual metrics such as accuracy, F1-score, recall, and specificity.

*Accuracy*: It is the ratio of the number of samples correctly classified by the model to the total number of samples. It is a widely used metric to measure overall performance.

$$Accuracy = \frac{TP + TN}{TP + TN + FP + FN}.$$

Here, TP (True Positive), TN (True Negative), FP (False Positive), and FN (False Negative) refer to the classification results.

*Precision*: Precision quantifies the proportion of predicted positives that are correct:

$$Precision = \frac{TP}{TP + FP}.$$

*Recall*: Refers to the rate at which the positive class is correctly predicted. It measures how well the model captures positive classes.

$$Recall = \frac{TP}{TP + FN}.$$

*Specificity*: Refers to the rate at which the negative class is correctly predicted. It measures how well the model discriminates negative classes.

**Table 2 Distribution of features across training, validation, and test subsets.**

| Set | Count | MS | FS | WA | LG | ToP | CGL | CN | FR | Exam entry status |
|---|---|---|---|---|---|---|---|---|---|---|
| Train | 827 | 24.75 | 23.19 | 23.43 | DC: 85<br>DD: 35<br>FF: 649<br>F: 58 | First: 366<br>Sec.: 461 | 1st: 245<br>2nd: 361<br>3rd: 139<br>4th: 82 | C1: 44<br>C2: 57<br>C3: 95<br>C4: 90<br>C5: 93<br>C6: 12<br>C7: 39<br>C8: 84<br>C9: 104<br>C10: 178<br>C11: 31 | 0.6491 | Took: 419<br>Didn't take: 408 |
| Val. | 156 | 20.80 | 19.76 | 20.03 | DC: 15<br>DD: 2<br>FF: 129<br>F: 10 | First: 66<br>Sec.: 90 | 1st: 46<br>2nd: 72<br>3rd: 25<br>4th: 13 | C1: 7<br>C2: 12<br>C3: 18<br>C4: 18<br>C5: 17<br>C6: 0<br>C7: 7<br>C8: 13<br>C9: 21<br>C10: 37<br>C11: 6 | 0.6635 | Took: 78<br>Didn't take: 78 |
| Test | 211 | 18.64 | 19.48 | 19.05 | DC: 20<br>DD: 4<br>FF: 170<br>F: 17 | First: 95<br>Sec.: 116 | 1st: 67<br>2nd: 88<br>3rd: 36<br>4th: 20 | C1: 10<br>C2: 18<br>C3: 26<br>C4: 22<br>C5: 22<br>C6: 0<br>C7: 11<br>C8: 19<br>C9: 30<br>C10: 44<br>C11: 9 | 0.6502 | Took: 108<br>Didn't take: 103 |

**Note:**
The distribution of key features across the training, validation, and test subsets. The table includes counts for categorical features such as 'letter grade', 'type of program', 'course grade level', 'course name', and 'exam entry status', as well as average values for numeric features like 'midterm score', 'final score', and 'weighted average score'.

$$Specificity = \frac{TN}{TN + FP}.$$

*F1-score*: It is a metric that measures the balance between precision and recall. It is used to evaluate the performance of the model, especially in imbalanced datasets.

$$F1\text{-}score = 2 * \frac{Precision * Recall}{Precision + Recall}.$$

We report this set of metrics because they align with the dual objectives of educational planning and student support. Precision captures the efficiency of resource allocation by indicating how many predicted attendees actually attend; recall captures service reliability and equity by indicating how many actual attendees are correctly identified for support. Specificity reflects the model's ability to avoid over-allocating resources to non-attendees under operational constraints. F1 summarizes the precision–recall trade-off when

**Table 3  Hyperparameter search space for model optimization.**

| No | Type | Options | Step size |
|---|---|---|---|
| 1 | Number of hidden layers | 1–4 | 1 |
| 2 | The number of neurons in each layer | 16–256 | 16 |
| 3 | Activation function of each layer | tanh, ReLU, sigmoid, LReLU | – |
| 4 | Dropout rates of each layer | 0–0.5 | 0.1 |
| 5 | Learning algorithm | Adam, Nadam, SGD, RMSProp, Adagrad | – |
| 6 | Learning rate | 0.01, 0.001, 0.0001 | – |

Note:
The hyperparameters and their respective ranges or options used during the model optimization process.

institutions value both minimizing under-provisioning (false negatives) and avoiding surplus (false positives). Reporting these metrics alongside accuracy enables administrators to choose decision thresholds consistent with policy priorities; in practice, threshold selection can shift the balance between precision and recall depending on whether missed attendees or over-provisioning is more costly. Although the target classes are approximately balanced in the dataset, these metrics remain informative for characterizing trade-offs relevant to deployment.

## Experimental setup

The training and evaluation of the model were carried out with a computer equipped with an Intel Core i7-10870H CPU, an NVIDIA GeForce RTX 2070 GPU, and 16 GB of RAM. Python programming language and various machine learning libraries were used for the development and training of the model. The main libraries used are as follows: TensorFlow (v2.9.0), scikit-learn (v1.1.1), NumPy (v1.22.4), Matplotlib (v3.5.2), and pandas (v1.4.3).

## Model and hyperparameter optimization

In this study, a feed-forward artificial neural network (ANN) was employed to predict student make-up exam participation. Although ANNs can learn highly non-linear relationships, their performance is sensitive to structural and training hyperparameters (*LeCun, Bengio & Hinton, 2015*). Exhaustive grid search over all plausible settings is computationally prohibitive—our full Cartesian grid contained $\approx 1.65 \times 10^{13}$ combinations—so random search was adopted (*Bergstra & Bengio, 2012*; *Claesen & De Moor, 2015*; *Feurer & Hutter, 2019*). Random search has been shown to cover more hyperparameter directions than grid search under an identical budget, is embarrassingly parallel, and avoids the design effort of hand-crafted Bayesian priors; preliminary trials confirmed that it achieved comparable validation performance to a small-scale Bayesian run at a fraction of the setup time. Further architectural details and the complete search grid are reported in Table 3 to facilitate reproducibility.

## RESULTS

In this study, an ANN was developed to predict students' participation in the make-up exam. Before model training, an exploratory correlation analysis was conducted to

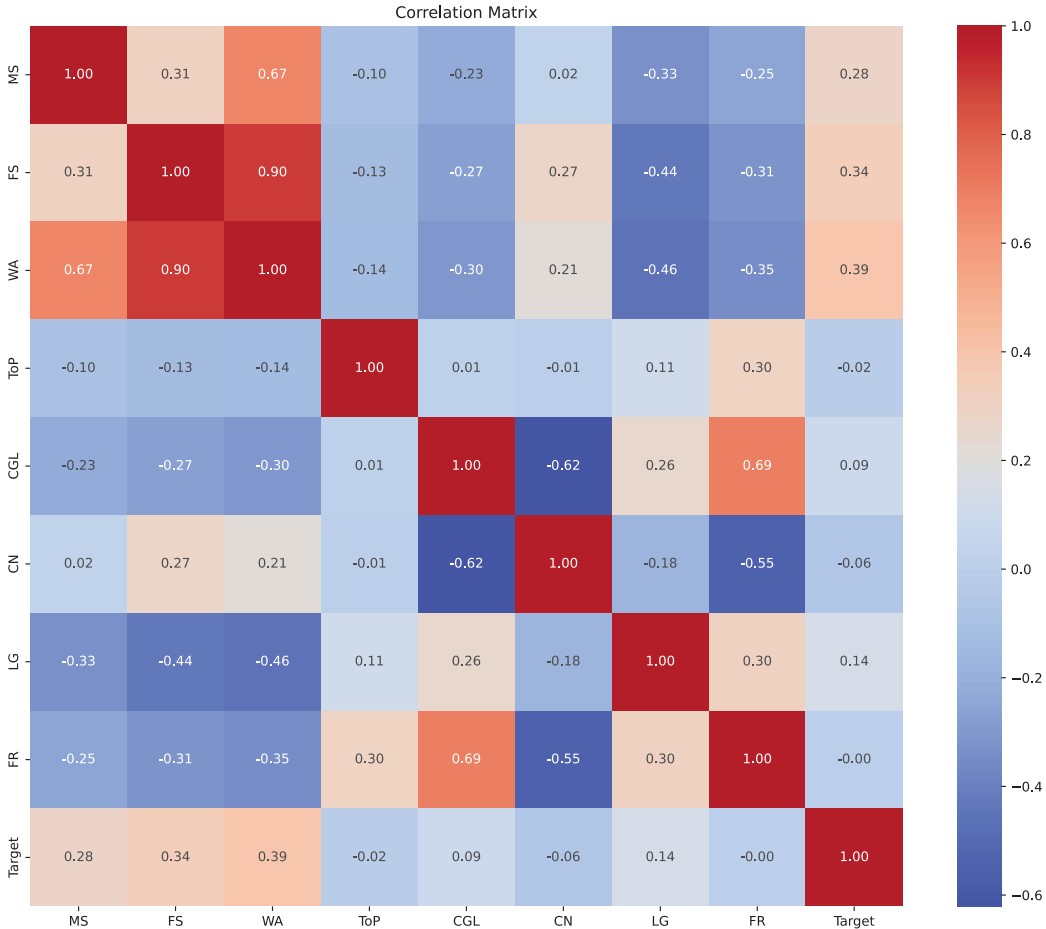

**Figure 2 Correlation matrix of variables.** Heatmap displaying the correlation coefficients between different variables. The values range from −1 to 1, where 1 indicates a perfect positive correlation, −1 indicates a perfect negative correlation, and 0 indicates no correlation. The variables are labeled along both axes.

characterize relationships among features and the target. While this analysis did not inform feature selection or model fitting, it provides interpretive context and supports the justification of the chosen inputs.

## Correlation analysis

We examined pairwise correlations among numeric variables and one-hot encoded categorical variables and visualized the results as a heatmap in Fig. 2. For binary variables, Pearson correlation corresponds to the point-biserial coefficient. Because some categorical variables are represented *via* multiple one-hot columns, correlations involving these should be interpreted with care.

Key observations were as follows. Midterm and final scores were positively correlated ($r = 0.31$), indicating broadly parallel performance across assessments. As expected, the weighted average was strongly correlated with both midterm ($r = 0.67$) and final scores ($r = 0.90$), reflecting its construction. Make-up participation showed positive correlations

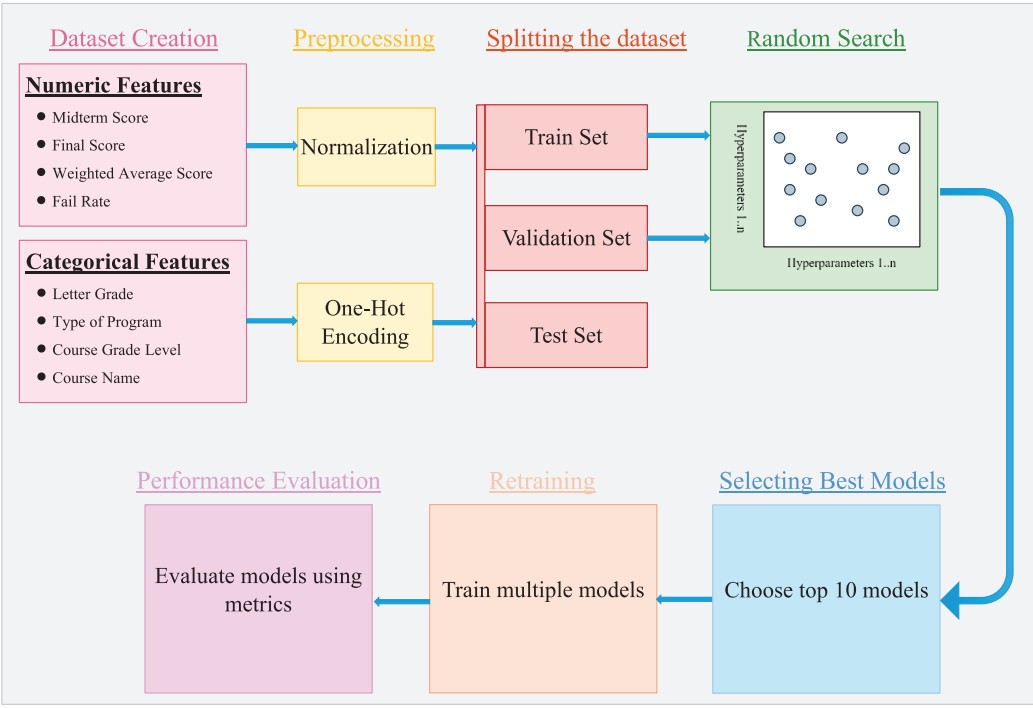

**Figure 3 Workflow for model training and evaluation.** Steps involved in the data preparation, model training, and evaluation process.

with midterm ($r = 0.28$) and final ($r = 0.34$) scores. Given that the dataset excludes students with passing grades (AA–CC), this pattern is consistent with students who are closer to the passing threshold within the failing range being more likely to attempt the make-up exam. The course-level fail rate was positively correlated with course grade level ($r = 0.69$), suggesting higher failure prevalence in upper-level courses. Letter grades were negatively correlated with midterm ($r = -0.33$) and final ($r = -0.44$) scores when encoded in the failing spectrum, as expected given grade construction. Overall, the observed associations suggest that score-related features and course context are informative for predicting make-up exam participation.

We emphasize that these correlations were used solely for exploratory insight and to check for obvious redundancies; the ANN was trained on the complete feature set, and no correlation-based pruning was applied. This choice avoids inadvertently discarding features whose relationships with the outcome may be non-linear or interaction-dependent.

## Performance evaluation

We optimized architectural and training hyperparameters *via* random search within the predefined space (Fig. 3 illustrates the model-building workflow; Fig. 4 details the selected ANN). The search evaluated 1,000 randomly sampled configurations. Each candidate was trained for up to 50 epochs with early stopping (patience = 10, monitor = validation loss)

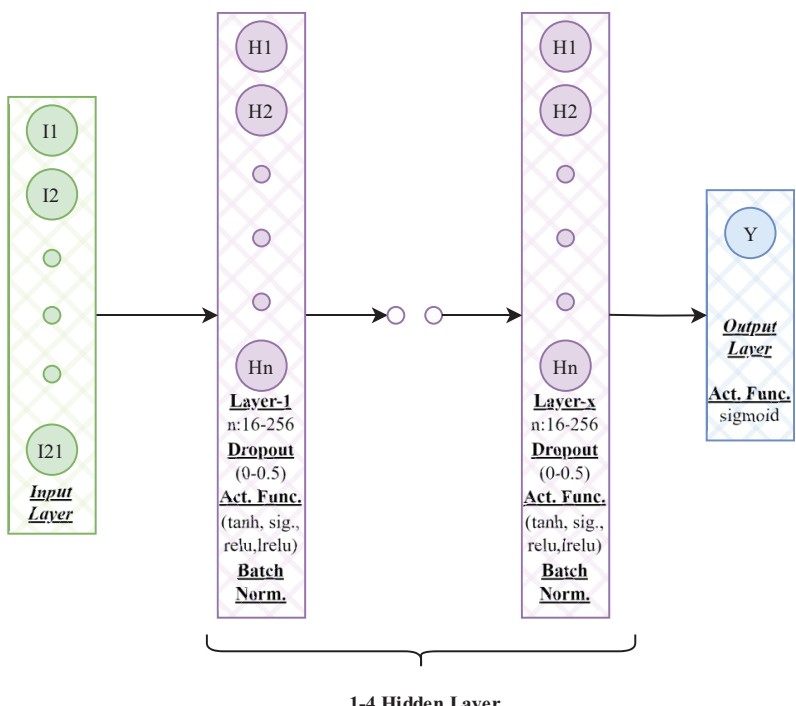

**1-4 Hidden Layer**

**Figure 4 Neural network architecture.** Diagram illustrating the architecture of a neural network. The network consists of an input layer, multiple hidden layers with activation functions (*e.g.*, sigmoid, tanh, relu, leaky relu), batch normalization, and dropout layers, followed by an output layer. The hidden layers are parameterized with varying neuron counts (*e.g.*, 16–256) and dropout rates (0–0.5).

to mitigate overfitting, and performance was assessed on the validation split. The top 10 configurations by validation loss were retrained 100 times each to assess stability. For each, the maximum and mean test accuracy were recorded. This procedure provided insight into hyperparameters that consistently yield strong performance. Summary results are reported in Table 4.

Across the top 10 configurations, the number of hidden layers took values of 1, 2, or 3 (encountered 4, 2, and 4 times, respectively); no four-layer models appeared in the top set. The highest single run test accuracy (87.68%) was obtained with a two-layer network (Model 1), suggesting that a relatively shallow architecture suffices for this task. Among optimizers, Root Mean Square Propagation (RMSProp) dominated (6 of the top 10), while Stochastic Gradient Descent (SGD) did not appear. Learning rates skewed higher within the explored range. ReLU activations were prevalent among the best performers (notably Model 1, Model 4, Model 5), consistent with stable convergence after normalization and one-hot encoding.

Considering both peak and average performance over 100 retrainings, Model 1 and Model 4 achieved the highest maximum test accuracy (87.68%). Model 5 yielded the highest mean test accuracy (84.08%). The proximity of mean and maximum accuracies across top models indicates stable generalization rather than reliance on fortuitous initializations.

**Table 4 Summary of candidate neural network architectures and hyperparameters with corresponding maximum and average accuracy across runs.**

| Model No | Hyperparameters (units-activation-dropout) | Max. accuracy (%) | Average accuracy (%) |
|---|---|---|---|
| Model-1 | Number of layers: 2<br>Layer 1: 160-ReLU-0<br>Layer 2: 224-tanh-0.1<br>Optimizer: RMSProp, LR: 0.01 | **87.68** | 83.49 |
| Model-2 | Number of layers: 3<br>Layer 1: 240-ReLU-0<br>Layer 2: 208-LReLU-0<br>Layer 3: 80-tanh-0.1<br>Optimizer: Adagrad, LR: 0.01 | 86.26 | 82.75 |
| Model-3 | Number of layers: 3<br>Layer 1: 144-tanh-0<br>Layer 2: 64-ReLU-0.1<br>Layer 3: 80-tanh-0<br>Optimizer: RMSProp, LR: 0.001 | 85.78 | 82.04 |
| Model-4 | Number of layers: 3<br>Layer 1: 96-ReLU-0.1<br>Layer 2: 224-ReLU-0.1<br>Layer 3: 192-sigmoid-0.1<br>Optimizer: Nadam, LR: 0.001 | **87.68** | 82.74 |
| Model-5 | Number of layers: 1<br>Layer 1: 256-ReLU-0.4<br>Optimizer: RMSProp, LR: 0.01 | 86.73 | **84.08** |
| Model-6 | Number of layers: 1<br>Layer 1: 144-tanh-0.4<br>Optimizer: RMSProp, LR: 0.01 | 84.36 | 82.56 |
| Model-7 | Number of layers: 1<br>Layer 1: 48-ReLU-0.3<br>Optimizer: Adam, LR: 0.01 | 84.36 | 82.10 |
| Model-8 | Number of layers: 3<br>Layer 1: 176-LReLU-0<br>Layer 2: 48-ReLU-0<br>Layer 3: 80-tanh-0.2<br>Optimizer: Adagrad, LR: 0.01 | 84.36 | 80.98 |
| Model-9 | Number of layers: 1<br>Layer 1: 240-ReLU-0.3<br>Optimizer: RMSProp, LR: 0.01 | 85.78 | 83.92 |
| Model-10 | Number of layers: 2<br>Layer 1: 240-tanh-0.1<br>Layer 2: 96-ReLU-0.3<br>Optimizer: RMSProp, LR: 0.01 | 85.78 | 82.68 |
| | | **Max: 87.68** | **Mean: 82.73** |

**Note:**
The maximum and average accuracy of various models, along with their corresponding hyperparameters. The text displayed in bold indicates the highest values.

For final selection, Model 1 was chosen based on its highest test accuracy (87.68%) and consistently strong performance across repetitions. Table 5 reports detailed test metrics for this model, and Fig. 5 presents the corresponding confusion matrices. On the predefined splits, accuracy values were 77.87% (training), 81.41% (validation), and 87.68% (test). Test

**Table 5 Classification performance metrics on the training, validation, test, and overall datasets.**

| Set | Accuracy | Precision | Recall | Specificity | F1-score |
|---|---|---|---|---|---|
| Train set | 77.87 | 82.60 | 71.36 | 84.56 | 76.57 |
| Validation set | 81.41 | 85.51 | 75.64 | 87.18 | 80.27 |
| Test set | 87.68 | 90.20 | 85.19 | 90.29 | 87.62 |
| Whole dataset | 80.07 | 84.43 | 74.38 | 85.91 | 79.09 |

**Note:**
The performance metrics of the model, including accuracy, precision, recall, specificity, and F1-score, across the training, validation, test subsets and entire dataset.

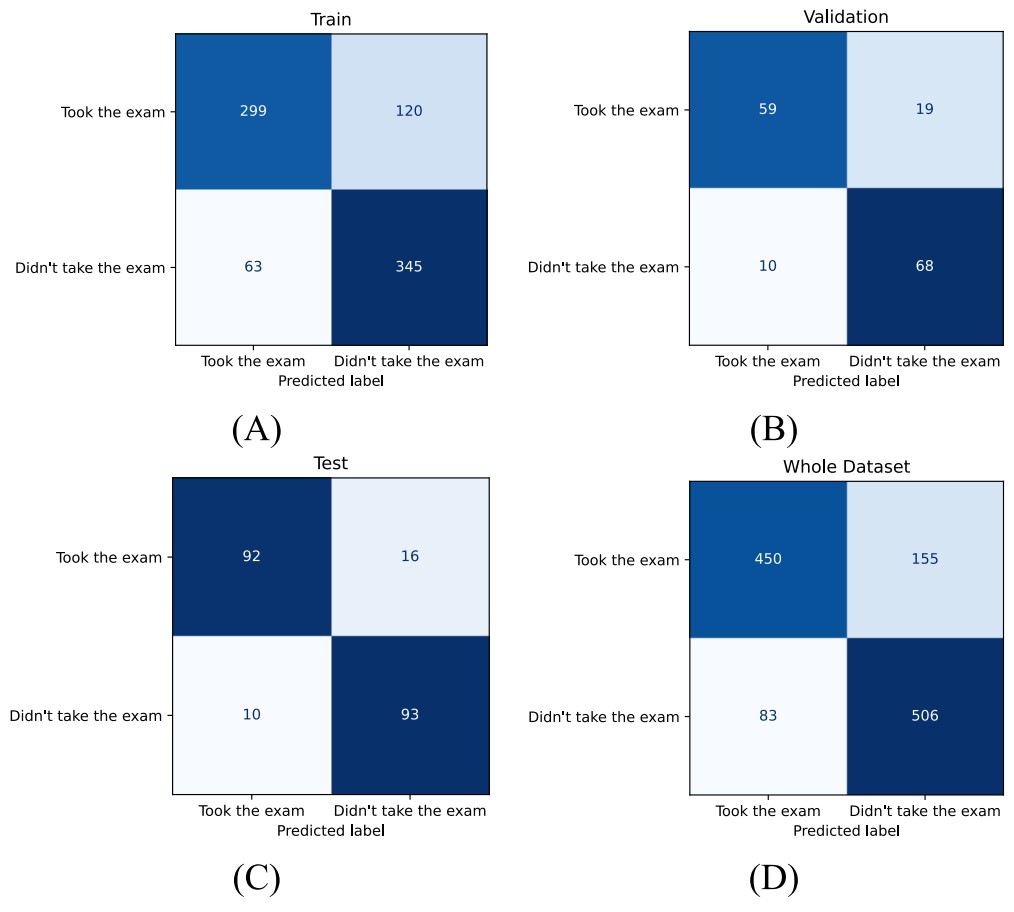

**Figure 5 Confusion matrices for model performance across different subsets.** Confusion matrices illustrating the performance of the model on four datasets: (A) training set, (B) validation set, (C) test set, and (D) entire dataset.

set precision, recall, specificity, and F1 were 90.20%, 85.19%, 90.29%, and 87.62%, respectively (Table 5). These results indicate robust out-of-sample performance on the held-out test set.

## Performance comparison

In this section, the performance of the model proposed in this study is compared with the findings of two reference studies, namely Study-1 and Study-2, and the contribution of the

study to the literature is evaluated. There are significant differences between the studies in terms of datasets, methods, and results. Study-1 (*Kiran et al., 2022*) and Study-2 (*Kiran, Siramkaya & Eşme, 2021*) used the same dataset with advanced features that are not easily available, such as the student's hometown distance to the university. This dataset allowed for more complex and detailed estimation. In contrast, in this study, a new dataset with more straightforward features is created, and the problem is approached more simply. This has a direct impact on the generalization ability and computational complexity of the model.

In Study-1, ANNs were trained with three different optimization methods: backpropagation, gray wolf optimization, and random weight initialization. The results revealed different strengths of each method. Backpropagation achieved the highest training accuracy, gray wolf optimization provided the best test accuracy (82.39%), and random weight initialization offered faster training times. This study highlighted the trade-off between training accuracy, generalization ability, and computational efficiency and showed that each method can provide advantages in different scenarios.

In Study-2, the same dataset was used to predict the number of students taking the make-up exam. The kernel-based Extreme Learning Machine (k-ELM) algorithm was used, and this algorithm was optimized using the ABC algorithm. This method enabled k-ELM to achieve higher classification accuracy than the baseline ELM on all datasets. The ABC algorithm played a critical role in optimizing parameters such as activation functions and the number of hidden-layer neurons and contributed significantly to the success of the model. Moreover, the use of multiple activation functions such as Sigmoid, Sinus, Hardlim, Tribas, Radbas, and TanSig increased the flexibility and adaptability of the k-ELM model.

In this study, a new dataset with more easily accessible simple features was created, and an ANN model was developed. The parameters of the model are optimized by a random search method. Although this method is not as sophisticated as ABC or gray wolf optimization, it provides a practical and effective solution for the current dataset. The highest test accuracy achieved was 87.68%. Given the simplicity of the dataset, this result is quite competitive and shows that a well-designed ANN model can be successful even with a less complex dataset. In addition, the dataset is publicly available, allowing researchers to improve performance.

## DISCUSSION AND LIMITATIONS

This study shows that an ANN trained on routinely available academic and course features can predict make-up exam participation with strong out-of-sample performance. Below, model behavior and deployment considerations are discussed, followed by an outline of limitations, validity threats, ethics, and future work.

### Generalization and validity checks

The higher test performance relative to training/validation can arise from benign factors such as early-stopping, regularization, or stochastic variation. To minimize the risk of leakage, fixed 0–1 rescaling (not estimated from data), deterministic encodings, and a single held-out split created prior to model selection were employed. Hyperparameters

were tuned using only the training/validation data; the test set was touched once for final reporting. The top configurations were also retrained 100 times to assess stability. These precautions make data leakage or overfitting unlikely, though it is acknowledged that random variation may favor the held-out split.

### Practical interpretation

Operational use requires selecting a decision threshold that balances precision and recall in line with institutional priorities. At the default operating point, the test profile is slightly precision-leaning yet balanced (precision = 90.20%, recall = 85.19%, specificity = 90.29%, F1 = 87.62%). In practice, false positives imply over-preparation (*e.g.*, surplus printing/invigilation), whereas false negatives risk under-provisioning for actual attendees. Institutions emphasizing reliability and equity can lower the threshold to increase recall; under tighter resource constraints, a higher threshold can increase precision and specificity. Figure 5's confusion matrices illustrate the error profile at the reported threshold.

### Operational considerations

Many institutions provide make-up exams assuming full attendance among eligible students. In the data, observed attendance among eligible students is roughly one-half. Using the reported precision–recall profile, planning based on predicted attendees with a chosen safety margin could substantially reduce article use and related provisioning *versus* a "print-for-all-eligible" baseline. Exact savings depend on local practices (*e.g.*, pages per pack, duplexing) and the safety margin. These are emphasized as secondary, operational benefits that complement the primary educational value of earlier, targeted student support.

### Integration and deployment

For real-world use, a privacy-preserving batch pipeline can generate predictions from routine student information system fields by applying the same fixed [0, 1] scaling and one-hot encoding, running inference, and then applying institutionally defined thresholds to produce actionable lists for scheduling. Practical deployment should handle unseen categories (*e.g.*, new course codes), schedule periodic recalibration or re-training, monitor subgroup performance, and include human-in-the-loop review with configurable safety margins. A lightweight Application Programming Interface (API) or secure file exchange is sufficient; the model and dataset are small enough to run on a CPU if no GPU is available.

### Limitations and external validity

A key limitation is the narrow feature scope, as the study uses only course-level and assessment features while excluding demographic and behavioral attributes. This improves feasibility and privacy but may limit predictive power and generalizability where additional factors meaningfully influence participation. The data derive from a single department at one university and a single semester, which constrains external validity. In addition, excluding students with passing grades (AA–CC)—a pragmatic choice to target the at-risk population—introduces selection bias; predictions are intended for students eligible for the

make-up exam and should not be extrapolated to the broader student body. Outcome prevalence and performance may vary with course mix, upper-level difficulty, and program type (first *vs.* second education), and institutional policies regarding make-up exams differ across universities and countries, affecting portability.

### Threats to validity and methodological trade-offs

We did not employ k-fold or nested cross-validation, opting instead for a single, clean held-out test benchmark and computationally tractable random search. The exploratory correlation analysis was used solely for interpretive context; correlation-based feature pruning was avoided to prevent losing non-linear or interaction effects.

### Ethical considerations

AI-driven predictions in education raise important ethical issues. Bias should be assessed and monitored to avoid disparate error rates across subgroups. Data privacy must be safeguarded; use should be purpose-limited and auditable. Transparency is essential so that stakeholders can understand how predictions are used, contest outcomes where appropriate, and intervene for students flagged as at risk. Thresholds and workflows should be calibrated to support students rather than penalize them, with human oversight for consequential decisions.

### Future work

We plan to (i) conduct external validation across departments and institutions, and temporal validation across semesters; (ii) extend comparisons to standard baseline algorithms and assess calibration; (iii) quantify operational impacts more precisely under different threshold policies and safety margins; and (iv) study deployment guardrails, including drift detection and subgroup fairness monitoring.

## CONCLUSIONS

In this study, an ANN model is developed to predict whether students will take the make-up exam or not by using a dataset obtained from the courses given at Selcuk University. Within the scope of the study, all processes, from the creation of the dataset to the training and evaluation of the model, are discussed in detail. The dataset was selected from courses at different academic levels and courses with varying degrees of difficulty to create a comprehensive sample that reflects students' academic performance and the general characteristics of the courses. In the data preprocessing stage, normalization was applied to numeric features, while categorical variables were converted into a numerical format using one-hot encoding. The dataset was divided into training, validation, and test subsets in a balanced way using the Stratified SPxy method, which preserved both the class distribution of the target variable and the diversity of input features. The hyperparameter optimization of the model was performed by random search, which evaluates random combinations in a large search space, and the best-performing hyperparameter combinations were determined. The results showed that the model achieved 87.68% accuracy on the test data. In addition, the model achieved high sensitivity (85.19%), specificity (90.29%), and F1-score (87.62%) on the test data, demonstrating its strong

generalization ability and applicability to real-world settings. Despite relying on simple, readily obtainable features, the model delivered competitive performance, suggesting that practical, deployable predictors can be built without sensitive or hard-to-collect data. These findings contribute to the educational data mining literature by demonstrating a portable approach tailored to institutional decision support.

## ACKNOWLEDGEMENTS

The author would like to thank Selcuk University for providing the data.

This study used Grammarly and ChatLLM Teams for language, style, and spelling corrections. These generative AI tools were employed to provide suggestions for linguistic consistency, spelling and punctuation checks, and to improve clarity and fluency. The final version and all content of the manuscript have been carefully reviewed and approved by the author.

### Funding

The author received no funding for this work.

### Competing Interests

The author declares that they have no competing interests.

### Author Contributions

- Ozcan Cataltas conceived and designed the experiments, performed the experiments, analyzed the data, performed the computation work, prepared figures and/or tables, authored or reviewed drafts of the article, and approved the final draft.

### Data Availability

The code and raw data are available in the Supplemental Files.

### Supplemental Information

Supplemental information for this article can be found online at http://dx.doi.org/10.7717/peerj-cs.3394#supplemental-information.

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
