# Peer review of "A data-driven artificial neural network approach to predict make-up exam participation in higher education"

_PeerJ Computer Science, doi:10.7717/peerj-cs.3394_

## Round 0.1 · original submission · Major Revisions

· Academic Editor

Major Revisions

Incorporate the suggestions of the reviewers.

**Language Note:** The review process has identified that the English language must be improved. PeerJ can provide language editing services - please contact us at [email protected] for pricing (be sure to provide your manuscript number and title). Alternatively, you should make your own arrangements to improve the language quality and provide details in your response letter. – PeerJ Staff

·

Basic reporting

The study is generally clear with appropriate raw data and figures. The following points should be noted;

- While the English is mostly understandable, a professional language editing service is recommended to improve fluency, especially in the Introduction and (e.g., lines 15-33 and 391-410) for the Discussion sections.

-There is some redundancy in the literature review (e.g., similar ideas repeated in lines 45-62).

- Summarizing Tables 1 and 2 with short highlights in the Results section would improve clarity

- The abstract is informative but would benefit from clearer articulation of the novelty and broader impact.

Experimental design

The research question is relevant, well-defined, and addresses a specific challenge in Turkish higher education. The model used is designed with practical considerations in mind, using easily available academic features. Appropriate preprocessing techniques (normalization, one-hot coding) and a justified dataset partitioning strategy (Stratif SPxy) are used in the study. Hyperparameter optimization via random search in a large parameter space is adequately explained and implemented. However, the following points should be noted.

-The choice to use only course-level and exam performance features may limit the broader applicability of the model. This should be discussed more critically.

-The rationale for excluding students with passing grades from the dataset is understandable, but it introduces a selection bias. This limitation should be explicitly acknowledged in the Discussion or Limitations section.

-The ANN architecture is explained, but the decision process for selecting the final model is somewhat buried in the narrative. A clearer summary (perhaps a diagram or table summarizing the decision points) would improve understanding.

Validity of the findings

-The results are clearly presented and supported by statistical metrics (accuracy, F1, sensitivity, specificity).

-The performance of the model on the test set (87.68% accuracy, 87.62% F1) indicates good generalizability and is competitive with previous studies.

-The comparative analysis with the two previous studies is well integrated and adds scientific value.

--The interpretation of the higher test performance compared to training/validation is attributed to generalization, but this could also indicate potential data leakage or over-fitting. Please comment on this in the Discussion section.

-The correlation analysis (Figure 2) is useful but does not directly contribute to the model; its relevance should be briefly justified.

--While the authors claim environmental benefits (line 30), this argument seems somewhat tangential to the main scope. It can be removed or minimized unless supported by more substantial evidence.

Additional comments

Dear Author/Authors,

This paper addresses a unique and practical problem in educational planning using a well-implemented artificial neural network model. The focus on using accessible features for prediction is commendable and makes this approach applicable for real-world applications in universities in Türkiye. Besides, the following points should be noted.

-Improve the English expression of the paper and eliminate unnecessary repetitions.

-Discuss potential limitations more clearly (e.g., generalizability to other universities or countries, impact of excluding successful students).

-Consider briefly discussing the ethical implications of using AI in student exam prediction systems.
This increases reproducibility and potential for future research.

Best regards

·

Basic reporting

-

Experimental design

-

Validity of the findings

-

Additional comments

In lines 36-42, you addressed the topic of artificial intelligence without mentioning generative artificial intelligence, artificial general intelligence, and superintelligence. I suggest you briefly discuss these terms and explain how your research relates to any of them.

Please provide the practical and theoretical importance and limitations of the study.

Reviewer 3 ·

Basic reporting

See detailed comments

Experimental design

See detailed comments

Validity of the findings

See detailed comments

Additional comments

In this article, the authors propose a machine learning-based approach using ANNs to predict students' participation in make-up exams in the Turkish educational system. The authors highlight the potential environmental and resource-saving benefits of their model, while also showing that it provides competitive performance even with a simpler dataset. However, the authors' work still needs several improvements as follows:
1. The introduction lacks a clear justification for why predicting make-up exam participation is important beyond logistical concerns. Expanding on how such predictions could enhance educational outcomes or student support systems would give the study a more profound significance.,

2. The dataset used for training the model is not fully detailed. The authors should provide more information on the data collection process, the potential biases within the dataset, and whether it is representative of other educational systems or specific to Turkish universities.

3. While the paper uses several machine learning techniques, it doesn't offer a comparison of the model's performance with other commonly used algorithms for similar tasks. Including comparisons with methods like Random Forests or Support Vector Machines would add depth to the evaluation.

4. The results section provides accuracy, sensitivity, and specificity, but does not discuss the potential trade-offs between these metrics. Given the practical application of the model, a deeper discussion of precision-recall trade-offs and false positives/negatives would strengthen the results.

5. The literature should be incorporated with more latest works, such as "student-performulator: student academic performance using hybrid deep neural network," "analysis of enrollment criteria in secondary schools using machine learning and data mining approach," "deep neural network-based feature encoding for automated health monitoring using large ai models in online communication systems," and so on.

6. The discussion section does not sufficiently explain why the random search method for hyperparameter optimization was chosen. A more in-depth comparison of hyperparameter optimization methods (such as grid search or Bayesian optimization) would help readers understand the choice and its implications for the model's performance

7. As the authors mention that normalization was applied, they do not explain how it affects the model's learning process. A more detailed explanation of the effect of normalization and one-hot encoding on the results would provide a clearer understanding of the data preparation steps.

8. The paper mentions that "environmental benefits" could be achieved through the use of this model, but there is no quantitative analysis or discussion on how these benefits would be realized in practice.

9. The model’s generalizability is discussed in terms of test data, but there is no mention of how the model would perform on entirely new data, outside of the current university.

10. The choice of metrics is standard but lacks further explanation in the context of educational systems. The authors should justify why these particular performance metrics were selected and how they align with the goals of predicting student behavior in education systems.

11. There is a lack of detail on how the model could be integrated into existing university administrative systems. Discussing the potential challenges and technical considerations of deploying the model in real-time or operational environments would increase the practical value of the paper.

12. The conclusion lacks specific suggestions for future work. Expanding on areas such as exploring other factors affecting make-up exam participation or testing the model on larger datasets would provide a clearer direction for future research and improvements

Reviewer 4 ·

Basic reporting

The manuscript is clearly written, professionally formatted, and logically structured.
The abstract and introduction provide an appropriate context and highlight the motivation for the study.
The manuscript references relevant literature, with good coverage of foundational and recent works in educational data mining and ANN applications.
Figures and tables are appropriately labeled, well-presented, and informative.

There are some grammatical and typographical issues (e.g., “accuracy values are 77.87% in the training set…” could be more formally presented)

Experimental design

The study clearly defines the problem and presents a structured approach for dataset preparation, model development, hyperparameter tuning, and evaluation.
The dataset is reasonably sized (1,194 samples) and includes relevant categorical and numerical features.
While hyperparameter tuning was thorough, no cross-validation was used to reinforce findings.

Validity of the findings

The paper does well to benchmark the model against prior studies using the same problem but with more complex features.
The reported metrics (accuracy, precision, recall, F1 score, specificity) support the claimed performance of the model.

Additional comments

This is a good contribution towards educational data mining, offering practical modeling insights and a reproducible methodology. Minor revisions in language, justification of findings, and broader generalizability considerations would enhance the manuscript's impact and robustness.

---

## Round 0.2 · Minor Revisions

· Academic Editor

Minor Revisions

Incorporate the comments of the reviewers.

·

Basic reporting

The manuscript is clearly written in professional English and follows a logical academic structure. The introduction provides adequate background and situates the study within existing literature. Current and relevant references include direct comparisons to related artificial neural network (ANN) and optimization approaches. Well-presented figures and tables cover confusion matrices, workflow, ANN architecture, dataset description, hyperparameter search, and performance metrics. The raw dataset is also provided and described in detail to ensure transparency.

Experimental design

The study represents original primary research within the aims and scope of PeerJ Computer Science. The research question is well defined: predicting student participation in make-up exams using routinely available academic and course features. The methodology is rigorous, with a clear description of dataset preparation, preprocessing, and sampling strategy (SPxy stratified splitting). The ANN model development is systematic, with hyperparameter optimization via random search over 1000 configurations. Methods are described in sufficient detail for reproducibility, supported by workflow diagrams and parameter tables.

Validity of the findings

Results are statistically sound and robust. Test set performance (87.68% accuracy, 87.62% F1-score, 90.20% precision, 85.19% recall, 90.29% specificity) demonstrates strong generalization. Comparative analyses with prior studies highlight the advantages of using a simpler, more accessible dataset while still achieving competitive performance. Limitations are transparently acknowledged (single-institution data, selection bias due to exclusion of passing students, lack of cross-validation). Ethical considerations (fairness, privacy, bias monitoring) are explicitly discussed. Conclusions are consistent with the research objectives and supported by the results.

Additional comments

The revisions have substantially improved the manuscript, particularly in dataset description, methodological transparency, and performance reporting.

The addition of confusion matrices, expanded methodological details, and clear presentation of hyperparameter optimization strengthen the work.

The discussion of limitations and ethical implications is well articulated and adds depth to the study.

Minor editorial issues remain (language polishing and figure/table consistency), but these can be addressed during the production stage.

Reviewer 4 ·

Basic reporting

no comment.

Experimental design

no comment.

Validity of the findings

no comment.

Additional comments

The authors have addressed all the comments.

---

## Round 0.3 · accepted · Accept

· Academic Editor

Accept

The paper can be accepted.

·

Basic reporting

Language and structure: The text is written in clear, fluent, and technically correct English. The introduction is well-grounded in the literature, and the context of the article is clearly outlined.

Literature coverage: Relevant recent studies (ANN, BiLSTM, k-ELM, GWO, etc.) are cited, and the current study's place within this literature is clearly stated.

Figures and tables: In this revision, figures for elements such as the methodology flow, model architecture, confusion matrix, and hyperparameter search strategy have been clearly added/improved. This appears to address the "clarity" criticisms raised in the previous round.

Data sharing: Raw data and methodology transparency have been ensured; the dataset structure, sample numbers, and class distributions are presented in detail.

Experimental design

Study Originality: The research question is clearly defined (predicting makeup exam participation using an ANN model). The study topic is appropriate for the journal's scope and targets a problem that has been studied only a limited number of times in the previous literature.

Methodological Clarity: Data preprocessing (normalization, one-hot encoding), SPxy sampling, and the training/validation/testing distinction are explained in detail. The rationale for choosing the random search strategy and the hyperparameter space is detailed.

Technical Standards: The training procedure, hardware information, used libraries, and performance metrics (accuracy, precision, recall, specificity, F1) are clearly stated.

Ethical Compliance: Data is anonymized, student personal data is not used; ethical compliance is properly addressed.

Validity of the findings

Support for the results: Results (accuracy: 87.68%, F1: 87.62%) are presented in a statistically robust manner; performance comparisons are made with studies in the literature (e.g., GWO and ABC optimization with k-ELM).

Data/analysis robustness: Data balance and sampling strategy are explained with respect to generalizability; overfitting/leakage risks are discussed.

Limitations and ethics: Data from a single institution, lack of cross-validation, and potential selection bias are clearly stated. These issues are honestly addressed in the discussion section.

Previous reviewer suggestions: All important comments have been addressed, including figure additions, methodology expansions, and strengthening of the discussion section.

Additional comments

Dear Editor,

The authors carefully considered all comments received from the previous round and made significant methodological and presentational improvements. The article now meets the journal's publication criteria. Only minor language/format adjustments were required during the production phase. I recommend acceptance for publication.

Best regards

Reviewer 3 ·

Basic reporting

no more comments

Experimental design

no more comments.

Validity of the findings

no more comments.

Additional comments

no more comments.